# Assessing the Effect of the Chinese River Chief Policy for Water Pollution Control under Uncertainty—Using Chaohu Lake as a Case

**DOI:** 10.3390/ijerph17093103

**Published:** 2020-04-29

**Authors:** Xia Xu, Fengping Wu, Lina Zhang, Xin Gao

**Affiliations:** 1Business School of Hohai University, Nanjing 21100, China; 170212070004@hhu.edu.cn; 2Decision and Planning Institute, Business School of Hohai University, Nanjing 21100, China; 3Business Administration School of Hohai University, Changzhou 213022, China; 20191001@hhu.edu.cn

**Keywords:** River Chief Policy, water pollution control effect, stochastic differential game, random interference coefficient, rewarding excellence and punishing inferiority coefficient

## Abstract

The River Chief Policy (RCP) is an innovative water resource management system in China aimed at managing water pollution and improving water quality. Though the RCP has been piloted in some river basins of China, few scholars have studied the effects of the policy. We built a differential game model under random interference factors to compare the water pollution in Chaohu Lake under the RCP and without the RCP, and we explored the conditions to ensure the effectiveness of the RCP. The results showed that: (1) The average effect of water pollution control under the RCP was greater than under non-RCP; (2) the higher the rewarding excellence and punishing inferiority coefficient (θ) was, the better the water pollution control effect under the RCP; (3) the greater the random interference coefficient (σ) and rewarding excellence and punishing inferiority coefficient (θ) were, the bigger the fluctuation of the water pollution control effect was; (4) when using the stochastic differential game, when σ≤0.0403, θ≥0.0063, or σ>0.0403, θ≥0.268, the RCP must be effective for water pollution control. Therefore, we can theoretically adjust the rewarding excellence and punishing inferiority coefficient (θ) and the random interference coefficient (σ) to ensure the effective implementation of the RCP and achieve the purpose of water pollution control.

## 1. Introduction

Water pollution is one of the most pressing environmental issues in many parts of the world [1,2,3,4,5,6]. The failure to meet the basic needs for safe water is one of the great tragedies of our age, and billions of people have suffered for it [7,8]. Water pollution control is one of the most important tasks for governments, and attempts have been undertaken by many countries around the world to manage the problem. For example, many European countries have unified their policy frameworks to centralize environmental supervision powers in order to solve the problem of water pollution [9,10]. Unfortunately, the concentration of environmental policies has had no effect on solving the problem of water pollution due to the strategy’s inability to maximize social benefits [10,11]. The United States adopted an integrated-decentralized management model to control water pollution [12], and empirical evidence has suggested that the free riding cost of a decentralized water environment policy is very high but it can improve regulatory efficiency [13,14]. Due to weak institutional structure, decentralized water policies in developing nations are less successful than in the United States [10]. For example, Brazil’s water pollution control policy has caused a large number of cross-border negative spillovers due to a lack of cooperation between local governments [13]; India’s poor enforcement of water pollution regulations might explain why these laws have had no measurable benefits [15].

In China, there are also many system constraints for water pollution control, including the complex and fragmented institutional arrangements, a lack of penalties or incentives for governments’ officials, and complex water administration [10,16,17]. This has resulted in ineffective pollution control measures across all levels of government in China. It is urgent for China to design and implement a new policy in order to control water pollution. On the 28th meeting of China’s Leading Group Comprehensive Deepening Reform, the River Chiefs Policy (RCP) was adopted and used to control water pollution in rivers and lakes [18].

The RCP involves appointing the leading local officials as “He Zhang” (river chiefs) for a specific river course, and it puts them in charge of the water conservation and management within their jurisdiction [3,10]. In addition, according to the water quality at the intersection of the administrative region, the superior government rewards or punishes the local officials (river chiefs) who are responsible for the region; this is called “the rewarding excellence and punishing inferiority” system [19,20]. Specifically, each body of water (divided by each administrative area) is managed by special local officials, and they are assessed by their superiors based on the improvement of water quality.

If the local government’s actual water pollution control effect meets the national standard, the state rewards the local government’s river chief, with such benefits as promotion; otherwise, if the local government’s actual water pollution control effect does not meet the national standard, the state punishes the local government’s river chief, like by demotion or criticism in public. The reason why “the rewarding excellence and punishing inferiority” system can be implemented in the RCP is that the central government of China has absolute authority over local governments, and the water quality is linked with a local official’s promotion or dismissal [10,19,20]. Therefore, this environmental policy provides a strong incentive for local officials (river chiefs) to control water pollution [10,19,20]. The RCP and the system of rewarding excellence and punishing inferiority comprise the creative institutional arrangement put forward by the Chinese government to control water pollution, which shows the determination of the Chinese government to deal with water pollution.

Since this is an attempt at policy to control water pollution in China, many scholars have studied the RCP from many different perspectives. Wang and Cai [21] analyzed the advantages and disadvantages of the RCP in regards to transaction costs, property rights, and institutional change theory. Zhan and Xiong [22] expounded the issues of the RCP for water pollution control and put forward three paths: the “rule of law, rule of virtue, and autonomy,” to optimize the RCP. Certain scholars [23,24,25,26] have focused on the history, the function, and the development of the RCP in China. Other scholars [10,27] have studied the changes of some water quality indicators under the RCP, and they have provided policy suggestions. Scholars [28,29,30] have also studied the RCP’s supervisory methods (technology, media, and the masses), assessment indicators (detect, task, and region), and assessment methods (government, profession, and society). These studies can help us understand the what is the RCP is, its path to governance, and how to assess it; however, there is a lack in understanding of how to evaluate and analyze the main coefficients that ensure effectiveness of the RCP from a quantitative perspective. Therefore, in this paper, through quantitative analysis, we can obtain more objective and accurate measurement results of the RCP, and we can take corresponding measures to ensure the effect of the RCP.

In addition, the effect of water pollution control (the reduction of water pollutions) presents uncertainties due to stochasticity, such as changes in hydrological processes, illegally discharged sewage, and sudden contaminant leakages [31,32]. Actually, water pollution control must be disturbed by random interference factors, and then the effect of water pollution control is disturbed [33,34,35,36]. While many scholars have studied the problems related to water pollution control, they have failed to consider the interference of random factors (stochastic events) [33,37,38,39,40,41,42]. Therefore, this paper assesses the integration of random interference factors in evaluating the effectiveness of water pollution control, both in the presence and absence of the RCP, and then it evaluates the RCP’s results more accurately.

Based on the above analysis, we found that there were two defects in the current studies evaluating the effect of the RCP on water pollution control: (1) There has been no quantitative study regarding the RCP’s water pollution control effect, nor have there been any analyses of the main coefficients to ensure the effect of the RCP; (2) the interference of random factors has been ignored, which could influence the water pollution control effect and produce a certain error related to the RCP’s effect. Therefore, in order to fix these research gaps, this paper chose Chaohu Lake as a research case. First, through the construction of a stochastic differential game model, we analyzed the effect of water pollution control when implementing the RCP and not implementing the RCP; secondly, we analyzed the main influencing coefficients that affect the effective implementation of the RCP; and finally, we obtained the specific conditions to ensure effective implementation of the RCP. In this paper, the stochastic differential game model had a considered random interference coefficient, and the equilibrium value was the interval number constructed by expectation and variance; through the, the influence of random factors on water pollution could be analyzed [43,44,45,46]. It is very important to evaluate the effect of RCP or other water pollution control measures around the world. Based on the stochastic differential game, we compared the RCP and non-RCP’s governance results in order to measure and ensure the effectiveness of the RCP policy more accurately.

This paper makes the following contributions: (1) It judges whether the implementation of the RCP is effective for water pollution control; (2) the main influencing factors of the RCP are theoretically discussed in order to determine the conditions to ensure the effectiveness of the RCP; (3) if the RCP is effective, then as a decentralization strategy for water pollution control, it can be used by many developing countries that are similar to China’s system. In addition, it will be valuable for the assessment of any country’s water pollution control policy to integrate random factors into the specific analysis of the water pollution control effect. Therefore, it has universal applicability to consider random interference factors in evaluating the water pollution control policy.

The rest of this paper is arranged as follows: In Section 2, we describe the materials and data sources. In Section 3, we introduce the methods. In Section 4, we provide the main results. In Section 5, we provide the main discussion. Finally, in Section 6, we present the conclusions.

## 2. Study Area and Data Sources

### 2.1. Study Area

Chaohu Lake is one of the five largest freshwater lakes in China, and it is located on the flood plain between the Yangtze River and the Huaihe River in the Anhui Province of eastern China [47]. More than 9.1 million people live in the Chaohu Lake basin, and it is a rapidly developing economic region with a number of important industries [48]. Chaohu Lake is a typical shallow lake with an average depth of 3 m and a surface area of 780 km^2^. The lake is located between 31°25′28″ and 31°43′28″ north latitude, as well as between 117°16′54″ and 117°51′46″ east longitude [48,49]. The detailed location of the Chaohu Lake basin system map is displayed in Figure 1.

Chaohu Lake is divided into east and west lakes, flowing from west to east. Water in the Chaohu Lake mainly comes from eight main inflowing rivers (R1–R8 in Figure 2) [48,50], which are the Dianpu, Shiwuli, Pai, Fengle, Baishishan, Zhao, Shuangqiao, and Zhegao rivers. Yuxi River (R9) is the only outlet, and water in Chaohu Lake flows to the Yangtze River from it. In Chaohu Lake, there are two important sampling points for water quality monitoring (S1 and S2). S1 is in the west center of Chaohu Lake and S2 is in the east center of Chaohu Lake [51]. Through the water quality monitoring of these two sampling points, the pollution control situation of Chaohu Lake water body can be determined [51].

According to the monthly report of Ministry of Ecological Environment of the People’s Republic of China [52], chemical oxygen demand (COD) has become the main pollutant in Chaohu Lake. COD (chemical oxygen demand) is a synthetical indicator that represents the degree of organic pollution in water [10]. Therefore, the change of COD in Chaohu Lake was taken as a case to judge whether the RCP is effective. The RCP was implemented in Chaohu Lake in September, 2017; we compiled the COD data of 2018 [53].

### 2.2. Data Sources

The water quality weekly data of COD in 2018 were collected from the website of Ministry of Ecology and Environment of People’s Republic of China and then averaged and counted. They are listed in Table A1 [51]. The limit values of the water quality ($Q_{3}$) were collected from the Environmental Quality Standards for Surface Water GB3838-2002, and they are listed in Table A2 [54]. The water qualification rate $\;\left(\tau \right)$ was collected from the website of Ministry of Ecology and Environment of People’s Republic of China [51]. The influence coefficient of the government’s efforts on water pollution control (α), the discount rate (γ), the decay rate of the water pollution control effect (δ), the unit cost coefficient of water pollution control (μ), the impact coefficient of water pollution treatment effect on the total revenue (β), and the rewarding excellence and punishing inferiority coefficient (θ) were derived from the relevant literature and then calculated by basic data [37,38,55,56,57,58].

## 3. Methods

### 3.1. Research Framework

Using the stochastic differential game [36,39], we compared the water pollution control effect between the unimplemented RCP and the implemented RCP under the interference of random factors, and then analyzed then main coefficients to ensure the RCP’s effect. The main ideas of this paper are as follows: First, we used a stochastic differential game to measure the expected value and variance of the water pollution control effect, and we ascertained the water pollution control effect’s upper and lower limit functions constructed by expectation and variance; secondly, the value of different coefficients in the model were determined; thirdly, we compared the interval range of the water pollution control effect between the unimplemented RCP and the implemented RCP, and then we analyzed the main influencing coefficients and how the main influencing coefficients in the model affected the water pollution control of the RCP; finally, we obtained specific conditions to ensure the effectiveness of the RCP’s implementation. The research framework is shown in Figure 2.

### 3.2. Model

#### 3.2.1. Hypothesis

**Hypothesis** **1.**
*The relationship between the efforts of local government in Chaohu Lake and the effect of water pollution control can be addressed by the stochastic differential equation (SDE) [36,59], which is shown in Equation (1):*
(1)$$dp\left(t\right)=\left[\alpha x\left(t\right)-\delta p\left(t\right)\right]d\left(t\right)+\sigma p\left(t\right)dw\left(t\right)$$
*where $\;p\left(t\right)$ represents the effectiveness of water pollution control at period t under the initial condition $\;p\left(0\right)=p_{0}\geq 0$. $dp\left(t\right)$ is the differential equation of $p\left(t\right)$. $x\left(t\right)$ represents the efforts of the local government in water pollution control at period t. α  indicates the influence coefficient of the government’s efforts on water pollution control. δ denotes the decay rate of the water pollution control effect. $w\left(t\right)\;$ represents the standard Wiener process denoted as the known the Brownian process. This represents the endless random movement of pollen particles suspended in liquid [60]. In mathematics, the Wiener process is a continuous time stochastic process [45,60]. We use σ to express randomness, which is the random interference coefficient.*


**Hypothesis** **2.**
*The costs for the pollution control measures of the local government mainly refer to the investments for environmental protection projects, quality water filter instruments, and so on, which comprise a quadratic function of the local government efforts [37,38]:*
(2)$$c\left(t\right)=\frac{\mu }{2}x^{2}\left(t\right)$$
*where $\;c\left(t\right)$ is the water pollution control costs of the local government at period t.  μ  denotes the unit cost coefficient of water pollution control in the local government.*


**Hypothesis** **3.**
*The total revenue of the local government after control water pollution can be expressed as [59]:*
(3)$$\pi \left(t\right)=\eta +\beta p\left(t\right)$$
*where $\;\pi \left(t\right)$ is the total revenue after treatment of water pollution at period t. β  denotes the influence coefficient of the water pollution control effect on the total revenue. η is a revenue constant.*


**Hypothesis** **4.**
*In the process of implementing the RCP, the Chinese government established an accountability system for local governments in water environmental governance [10,16]. Then, the rewarding excellence and punishing inferiority was added to the total revenue of the local government for water pollution control, and it can be defined as follows [37]:*
(4)$$\pi^{{}^{\circ}}\left(t\right)=\eta +\beta p\left(t\right)+\theta \left(x^{h}\left(t\right)\tau \left(t\right)-x_{0}^{h}\left(t_{0}\right)\tau_{0}\left(t_{0}\right)\right)$$
*where $\pi^{{}^{\circ}}\left(t\right)$ is the total revenue under the RCP at period t.  θ represents the coefficient for rewarding excellence and punishing inferiority. $x^{h}\left(t\right)$ represents the effort function of the local government in water pollution control under the RCP at period t. $\;\tau \left(t\right)\;$ denotes the water qualification rate at period t. $x_{0}^{h}\left(t_{0}\right)\;$ and $\tau_{0}\left(t_{0}\right)$ are the standard values of local government’s pollution control efforts and water qualification rate, respectively, which are set by the superior government to local governments.*


To clearly present the modeling, the notations are provided in the Appendix A (see Table A3). Additionally, in order to write conveniently without causing confusion, time *t* is omitted in the following section.

#### 3.2.2. Model Construction

(1)The interval value of the water pollution control effect without the RCP.

When the RCP is not implemented, the local government is not be evaluated by the superior government. The local government efforts for water pollution control are mainly aimed at maximizing the economy benefits within its jurisdiction. Therefore, based Equations (2) and (3), the objective function is formulated as:(5)$$Y_{1}=\max\;\;\;E\left\{\int_{0}^{t}e^{-\gamma t}\left[\pi -c\right]d\left(t\right)\right\}\;$$
where γ indicates the discount rate.

In order to make Equation (5) have a unique and continuous solution, we constructed a continuous and differentiable value function, $v\left(p\right),$ and this function satisfies the following Hamilton–Jacobi–Bellman (HJB) equation [36]:(6)$$\gamma v\left(p\right)=max\left\{\pi -c+v^{\prime }\left(p\right)\left(\alpha x-\delta p\right)+\frac{\delta^{2}}{2}v^{\prime \prime }\left(p\right)\right\}$$
where $\;v^{\prime }\left(p\right)\;\mathrm{and}\;v^{\prime \prime }\left(p\right)\;$ are the first and second derivatives of$\;v\left(p\right)\;$, respectively.

Based on Equation (6), we obtained the equilibrium value of the local government’s efforts on water pollution control, shown as Equation (7):(7)$$x^{*}=\frac{\alpha \beta }{\left(\gamma +\delta \right)}.\;$$

The stochastic differential equation regarding the water pollution control effect by incorporating Equation (7) into Equation (1) is shown as Equation (8):(8)$$dp\left(t\right)=\left[\Omega -\delta p\left(t\right)\right]d\left(t\right)+\sigma p\left(t\right)dw\left(t\right)\;,\Omega =\frac{\alpha^{2}\beta }{\mu \left(\gamma +\delta \right)}.\;$$

Finally, we rewrite Formula (8) as the$\;It\hat{o\;}$ form and solve it, and then we obtain the water pollution control effect’s expectation and variance under the unimplemented RCP, which are shown as Equations (9) and (10). The specific proof process is shown in the Appendix A.
(9)$$E\left(p\left(t\right)\right)=\frac{\Omega }{\delta }+e^{-\delta t}\left(p_{0}-\frac{\Omega }{\delta }\right)\;,\underset{t\to \infty }{\lim}E\left(p\left(t\right)\right)=\frac{\Omega }{\delta }=\frac{\alpha^{2}\beta }{\delta \mu \left(\gamma +\delta \right)}$$
(10)$$D\left(p\left(t\right)\right)=\frac{\sigma^{2}\left[\Omega -2\left(\Omega -\delta p_{0}\right)\right]e^{-\delta t}+\left(\Omega -2\delta p_{0}\right)e^{-2\delta t}}{2\delta^{2}},\underset{t\to \infty }{\lim}D\left(p\left(t\right)\right)=\frac{\sigma^{2}\Omega }{2\delta^{2}}=\frac{\sigma^{2}\alpha^{2}\beta }{2\delta^{2}\mu \left(\gamma +\delta \right)}.$$

Based on Equations (6) and (7), we can see that the water pollution control effect deviates from the expected value as a result of the random interference coefficient disturbance. The simulation diagram is shown in Figure A1.

From Figure A1, we can see the effect of water pollution control around its expected value fluctuates under different deserved periods. This implies that the value of the water pollution control effect is uncertainty. Therefore, the value of the water pollution control effect should be characterized as a confidence interval [61]. The water pollution control effect under a 95% confidence level is defined by:(11)$$E\left(p\left(t\right)\right)-1.96D\left(p\left(t\right)\right),E\left(p\left(t\right)\right)+1.96D\left(p\left(t\right)\right).\;$$

(2)The interval value of water pollution control effect with the RCP.

In the RCP, the superior government carries out environmental performance assessment for the local government, and it implements the strategy of rewarding excellence and punishing inferiority actions for the local government [10,27]. Therefore, the local government not only maximizes the benefits of the local region but also meets the environmental performance assessment from the superior government. Based on Equations (2) and (4), the objective function of the local government is formulated as follows:(12)$$Y_{2}=max\;\;\;E\left\{\int_{0}^{t}e^{-\gamma t}\left\{\pi^{{}^{\circ}}-c\right\}d\left(t\right)\right\}.$$

For the case of considering the RCP, the programs of water pollution control effect under a 95% confidence level can be established by dealing with the denominator (numerator) of the objective functions of Equations (6)–(10).
(13)$$E^{h}\left(p\left(t\right)\right)-1.96D^{h}\left(p\left(t\right)\right),E^{h}\left(p\left(t\right)\right)+1.96D^{h}\left(p\left(t\right)\right).\;$$

Despite the influence of random interference factors, the confidence interval (Equations (11) and (13)) contains most of the water pollution control effect values. Therefore, using the confidence interval to compare the water pollution control effect values of the two cases is a more accurate process that allows us to measure the RCP’s effect more accurately.

The main variables and the RCP’s description are provided in the Appendix A (see Table A4 and Table A5).

### 3.3. Definition of Parameters

(1)The decay rate of the water pollution control effect.

This can be expressed as follows [38,62]:(14)$$\delta =\frac{1}{t}ln\left(\frac{Q_{1}}{Q_{2}}\right)\;$$
where$\;Q_{1\;}\;\mathrm{and}\;Q_{2}\;$ denote the COD discharge of the upstream region (S1) and downstream region (S2) obtained from Table A1. *t* refers to the period.

(2)The influence coefficient of the government’s efforts on the effect of water pollution control.

α can be expressed as follows [33]:(15)$$\alpha =\left|\frac{Q_{1}-Q_{2}}{Q_{1}}\right|.\;$$

(3)The influence coefficient of the water pollution control effect on the total revenue.

β can be expressed as follows [58,63]:(16)$$\beta =\frac{B_{g}}{B_{t}}$$
where$\;B_{t\;}$ is the total revenue and $B_{g}\;$ is the green revenue.

(4)The rewarding excellence and punishing inferiority coefficient.

θ can be expressed as follows [56]:(17)$$\theta =\left\{\begin{array}{c} {\left|\frac{Q_{2}}{Q_{3}}-1\right|}^{w},\;Q_{2}\leq Q_{3}\;\;\\ {\left|\frac{Q_{3}}{Q_{2}}-1\right|}^{w},\;Q_{2}>Q_{3}\; \end{array}\right.$$
where w is an adjustment coefficient that reflects the degree of incentive$\;\left(w\epsilon \left[2,10\right]\right)\;$ [56]. When the value of w is higher, the value of θ is lower. This means that θ is a decreasing function. When w=2, θ reaches a maximum value, and when w=10, θ reaches a minimum value. $Q_{3}$ is the Environmental Quality Standards for Surface Water GB3838-2002 [54].

(5)The other coefficients.

$\sigma \in \left(0,1\right)$ is a probability regarding the possibility of random interference in water pollution treatment. With the increasing of random interference coefficient (σ), the external interference factors account for more, which means that water pollution control effect in Chaohu Lake is affected by more factors. There were no major random disturbances factors in Chaohu Lake in 2018 [64], such as extreme climate change or unexpected contaminant leakage; thus, we assume σ changes between 0 and 0.25 [5,65].

τ refers to the water qualification rate. According to the information provided by the Ministry of Ecology and Environment of People’s Republic of China [52], we were able to obtain a water qualification rate equal to 0.985.

μ refers to the unit cost coefficient of water pollution control. We assumed that the cost of sewage discharge in each region was the cost of pollution control [57]. According to the Standard Management Measures for Collection of Pollution Discharge Fees [55,57], we set μ with a value of 0.7. γ  denotes the discount rate and equals 0.005 [37,38].

### 3.4. The Criteria for Judging the Effectiveness of the RCP in Water Pollution Control

According to the stochastic differential game model, we found that the effect of water pollution control was disturbed by the random interference coefficient (σ) whether the RCP was implemented or not. Therefore, based on Equations (11) and (13), we can obtain the interval functions of the effect of water pollution control in the case of the RCP or not—these are called them the upper limit function and the lower limit function. The upper limit function represents the best situation of the water pollution control effect, and the lower limit function represents the worst situation of the water pollution control effect. Therefore, if the RCP’s lower limit is larger than the upper limit of the non-RCP, we can say that the RCP is effective. In order to more clearly express the upper and lower limit functions of water pollution control in the case of the RCP and the non-RCP, we use different symbols to express them, as shown in Table 1.

In order to judge whether the RCP is effective for water pollution control, we only need to judge the upper limit function of water pollution control effect without the RCP ($y_{2}$) and the lower limit function of water pollution control effect with the RCP ($y_{3}$). If $y_{3}>y_{2}$, this indicates that the RCP is effective for water pollution control; if $y_{3}<y_{2}$, this indicates that the RCP is invalid for water pollution control. If we cannot accurately compare $y_{2}$ and $y_{3}$, then this indicates that the effect of the RCP is related to the random interference coefficient (σ). When σ takes different values, the RCP is valid; when taking other values, it is invalid.

In this case, we needed to use the average value of the upper and lower limit functions as the basis for judgment, that is the average value of the upper and lower limit functions without the RCP are greater than the average value of the upper and lower limit functions with the RCP, which means that the RCP is invalid for water pollution control at the approximate rate; otherwise, it means that the RCP is effective. In this case, we need to continue to explore the impact of parameters on the RCP in order to find the appropriate path to promote the effectiveness of the RCP.

## 4. Results

### 4.1. The Value of Each Coefficient

Based on basic data (listed in Table A1 and Table A2) and Equations (14), (15), and (17), we obtained the values of δ, α, and θ, respectively. Other values of coefficients, such as β, τ, μ, γ, and σ, were obtained from related literature and documents. The values of all the coefficients are listed in Table 2. In addition, following the analysis of Section 3.3, θ refers to the rewarding excellence and punishing inferiority coefficient, and its value is different when implementing the RCP and not implementing the RCP; thus, it is an interval number. Similarly, σ is also an interval value.

### 4.2. Determination of Water Pollution Control Effect in the Two Situations

#### 4.2.1. The Numerical Expression of Water Pollution Control Effect’s Lower and Upper Limit Functions under the Unimplemented RCP

According to Section 3.4, we know that the lower and upper limit functions of the pollutant control effect in Chaohu Lake are$\;y_{1}$ and $y_{2}$, respectively. Putting the values (listed in Table 2) into Equation (11), we obtained a numerical expression of the lower and upper limit functions of COD in Chaohu Lake, which are listed in Table 3.

#### 4.2.2. The Numerical Expression of Water Pollution Control Effect’s Lower and Upper Limit Functions under the Implemented RCP

According to the expected value under the RCP, we found that the average value of the water pollution control effect was a linear increasing function of θ. Therefore, when the value of θ changed from 0.0063 to 0.363, the lower and upper functions generally showed an increasing trend. In order to better illustrate this, we randomly selected four values of θ to analyze the change of the water pollution control effect under the RCP. This showed that the higher of θ value, the greater of the lower limit of water pollution control effect. Similarly, with the increase of θ value, the upper limit of water pollution control effect also increased, which is shown in Figure 3A,B. Therefore, when θ=0.0063, we obtained the minimum lower and upper limit functions, and when θ=0.363, we obtained the maximum lower and upper limit functions. In addition, according to Section 3.4, we know the lower and upper limit functions of the pollutant control effects are represented by $y_{3}$, $y_{4}$, $y_{5},$ and $y_{6}\;$ when θ was equal to 0.0063 and 0.363. Putting the values (listed in Table 2) into Equation (13), we could obtain the numerical expressions of $y_{3}$, $y_{4}$, $y_{5},$ and $y_{6}$, which are listed in Table 4.

#### 4.2.3. Determination of Water Pollution Control Effect’s Average Value in the Two Situations

As the average value of water pollution control effect was a linear increasing function of θ, when θ=0.0063, the average value of the water pollution control effect was the minimum average value under the RCP. This means that we could choose the average value of the water pollution control effect under the RCP obtained from θ=0.0063 as a representative to compared with the average value of the water pollution control effect under the non-RCP. If the water pollution control effect’s mean value under θ=0.0063 was greater than the water pollution control effect’s mean value of the non-RCP, this indicates that the RCP was valid as a whole; otherwise, it was invalid as a whole. The results of the water pollution control effect’s mean value in the two situations are listed in Table 5.

#### 4.2.4. The Water Pollution Control Effect and the Effect’s Volatility of in the Two Cases

According to the numeric expressions of $y_{1}$,$\;y_{2}$,$\;y_{3}$,$\;y_{4}$,$\;y_{5},$ and$\;y_{6}$, which are listed in Table 3 and Table 4, we could obtain the results about the water pollution control effect’s point value and volatility when $\sigma \in \left(0,0.25\right)$. These are listed in Table 6 and Figure 4. The effect’s volatility was the water pollution control effect upper limit function minus the upper limit function; that is to say, it was the water pollution control effect upper limit function and lower limit function value’s gap; the bigger the gap, the bigger the volatility. According to Table 6 and Figure 4, the greater the random interference coefficient (σ), and the bigger the fluctuation of the water pollution control effect.

## 5. Discussion

### 5.1. Comparison of the Water Pollution Control Effect in Two Cases

According to the analysis of Section 3.4, if the upper limit numeric expression under the unimplemented RCP ($y_{2})$ was less than the lowest limit numeric expression under the implemented RCP ($y_{3}$), the RCP was effective; otherwise, the RCP was invalid. Based on the calculation results and Table 6 and Figure 4, $y_{2}$ and $y_{3}$ did not have a strict size relationship when $\sigma \in \left(0,0.25\right)$; that is to say, $y_{2}$ may have been larger than $y_{3}$ in some intervals, while $y_{2}$ was smaller than $y_{3}$ in other intervals. This means that the RCP is not effective in all cases.

Therefore, in order to further explore the effectiveness of the RCP, we compared the average value of the water pollution control effect under the condition of implementing the RCP and not implementing the RCP. Based on Section 4.2.3, in the case of the non-RCP, the average effect of water pollution treatment was represented by the arithmetic mean of the corresponding upper limit function ($y_{2}$) and the lower limit function ($y_{1}$); in the case of the RCP, the average effect of water pollution treatment was represented by the arithmetic mean of the corresponding upper limit function ($y_{3}$) and the lower limit function ($y_{4}$). According to this idea, when θ was equal to the minimum value (0.0063), the average effect of water pollution control with the RCP was 0.876, while the average effect of water pollution control without the RCP was 0.869. Therefore, through the comparison of the average effect of water pollution control in two situations, the RCP was still effective, that is to say, the implementation of the RCP can play a positive role in water pollution control.

### 5.2. The Effect of Rewarding Excellence and Punishing Inferiority Coefficient (θ) and Random Interference Coefficient (σ) on Water Pollution Control in Two Cases

According to the above analysis, when implementing the RCP, the rewarding excellence and punishing inferiority coefficient (θ) and random interference coefficient (σ) had an impact on the effect of water pollution control; when not implementing the RCP, only the random interference coefficient (σ) had an impact on the effect of water pollution control.

According to Table 6 and Figure 4, in the case of the implementation of the RCP, the effect of water pollution control increased with the increase of the coefficient of reward and punishment (θ) when$\;\sigma \in \left(0,0.25\right)$. According to Table 4 and Table 6, when θ equaled the minimum value of the interval (0.0063), the upper and lower limit functions of the water pollution control effect were $y_{3}$ and $y_{4},$ respectively, and $y_{4}$ was greater than $y_{3}$; when θ equaled the maximum value of the interval (0.363), the upper and lower limit functions of the water pollution control effect were $y_{5}$ and $y_{6},$ respectively, and $y_{6}$ was greater than $y_{5}$.

By comparing the size of $y_{4}$ and $y_{5}$, we found that $y_{5}$ was larger than $y_{4}$ at this time. Therefore, we could obtain the unequal relation of $y_{6}$ > $y_{5}$ > $y_{4}$ > $y_{3}$. This showed that, in the implementation of the RCP, the rewarding excellence and punishing inferiority coefficient (θ) had a positive impact on the effect of water pollution control. When the value of θ was larger, the effect of water pollution control was better; on the contrary, when the value of θ was smaller, the effect of water pollution control was worse. Comparatively, when the RCP was not implemented, the superior government was not able to supervise the local government. Therefore, in this case, there was no rewarding excellence and punishing inferiority coefficient (θ). Then, the reward and punishment coefficient (θ) had no effect on the effect of water pollution control. In addition to the rewarding excellence and punishing inferiority coefficient (θ), the random interference coefficient (σ) was also one of the main factors that affected the effect of water pollution control, and this factor had an impact on the effect of water pollution control, regardless of whether the RCP was implemented.

Based on Figure 4, no matter whether the RCP was implemented, we could easily find that the upper limit function of the water pollution control effect ($y_{2},y_{4},$ and $y_{6}$) increased with the increase of random interference coefficient (σ), while the lower limit function ($y_{1},\;y_{3}$ and $y_{5}$) decreased with the increase of random interference coefficient (σ). This showed that random interference coefficient (σ) had a positive effect on the upper limit function ($y_{2},y_{4},$ and $y_{6}$) of the water pollution control effect and a negative effect on the lower limit function ($y_{1},y_{3},$ and $y_{5}$). Due to this, when *σ*∈ (0,0.05), the upper and lower limit functions of the water pollution control effect were very close; therefore, the difference between them was relatively small. When σ∈ (0.05, 0.25), the difference between the upper and lower limit functions increased. This was because the upper limit function of the water pollution control effect is an increasing function, while the lower limit function is a decreasing function. Therefore, the difference between the upper and lower limit functions gradually expands.

### 5.3. The Analysis of the Main Influence Coefficients

Based on the analysis of Section 5.1, we know that the average effect of water pollution control under the RCP was better than that of not implementing the RCP; however, there were also invalid intervals. Based on Section 5.2, we know that the rewarding excellence and punishing inferiority coefficient (θ) and random interference coefficient (σ) could affect the effect of water pollution control. Therefore, in order to find out the range of parameters that could ensure the effectiveness of the RCP, we need to further analyze the rewarding excellence and punishing inferiority coefficient (θ) and the random interference coefficient (σ).

#### 5.3.1. The Effect of Random Interference Coefficient (σ) on Water Pollution Control in Two Cases

Based on Figure 4, we found that $y_{3}>y_{2}$ can only exist between 0 and 0.05. As $y_{2}$ and $y_{3}$ are two quadratic functions, in order to further determine the size of these two functions, we made $y_{2}$ = $y_{3}$, and we calculated that one of the intersection values was 0.0403, which belonged to the range of 0–0.05 (Figure 5). The other intersection value did not belong to the range 0–0.25; thus, it was omitted. Through calculation, we found that when the upper limit numeric expression under the unimplemented RCP ($y_{2})$ was less than the lower limit numeric expression under the implemented RCP ($y_{3}$) when 0<σ≤0.0403; and the upper limit numeric expression under the unimplemented RCP ($y_{2})$ was more than the lower limit numeric expression under the implemented RCP ($y_{3}$) when 0.0403<σ<0.25. This showed that when one only considers the impact of the random interference coefficient (σ) on the effect of water pollution control, in order to ensure the effectiveness of the RCP, the random interference coefficient (σ) should be between 0 and 0.0403.

The random interference factors that affect the effect of water pollution control and how to reduce the random interference in water pollution control are the key issues that we must pay attention to. Random interference factors come not only from natural factors but also from social and economic factors. In the natural field, abnormal changes of climate and hydrological characteristics change the effect of water pollution control [66,67]. At the same time, the abuse of agricultural fertilizer and the behavior of enterprises secretly discharging sewage (many enterprises secretly discharge their sewage directly into rivers and lakes after working hours) will also reduce the actual effect of water pollution control [36,68].

Therefore, we need to reduce the impact of random interference factors on the effect of water pollution control through human interventions in order to further ensure the effectiveness of the RCP. For climate anomalies in natural factors, these mainly affect water resources through temperature and precipitation [69]. At different temperatures, the evaporation rate of water changes, and different precipitation levels lead to different soil degradations, river flows, and aquifer infiltrations [67].

Therefore, we can increase the amount of vegetation in the reclamation area to alleviate temperature rises and soil erosion, and then we can reduce the impact of climate on water resources and the effect of water pollution control. For the random interference factors, we can reduce these random interference factors by reducing the amount of fertilization, managing the methods and timing of fertilization (which can reduce the loss of excessive nutrients), closing or reallocating polluting enterprises, cleaning streets (which can reduce the amount of pollution on the urban surface, which can be transferred to the receiving water body through surface runoff), etc., to reduce the effects of these random interference factors on the control of water pollution [66,68,70,71].

#### 5.3.2. The Effect of the Rewarding Excellence and Punishing Inferiority Coefficient (θ) on Water Pollution Control in Two Cases

As the rewarding excellence and punishing inferiority coefficient (θ) only affects the effect of water pollution control when the RCP is implemented, in order to ensure the effectiveness of the RCP, we only need to ensure that the upper limit function of the effect of water pollution control without the RCP ($y_{2}$) must be less than the minimum value of the lower limit function of the effect of water pollution control with the RCP ($y_{3}$). As $y_{3}$ is a monotonic subtractive function about σ, when σ obtains the maximum value of 0.25, $y_{3}$ can obtain the minimum value of $y_{3}^{\prime }$. Thus, we needed to compare the size of $y_{3}^{\prime }$ and $y_{2}$ at this time. Therefore, we made $y_{2}$ = $y_{3}^{\prime }$ and calculated that one intersection value of *θ* was 0.268, which belonged to the range of 0.0063–0.363 (Figure 6); the other intersection value of θ did not belong to the range; therefore, it was discarded. This means that for the RCP to be valid, the value of θ should be between 0.268 and 0.363.

The rewarding excellence and punishing inferiority coefficient (θ) means that if the local government’s water pollution treatment effect meets the national standard, the state rewards the local government’s river chief; otherwise, if the local government’s water pollution treatment effect does not meet the national standard, the state punishes the local government’s river chief. Therefore, for the effect of water pollution control, the efforts of the river chief in local governments are also a key factor.

In China, the efforts of the local government river chief to control the water pollution are influenced by incentive mechanisms and financial mechanisms [72]. The incentive mechanism mainly refers to the promotion mechanism of the river chief, and the effect of water pollution control is becoming more and more important in the promotion of local government officials. The effect of water pollution control on the promotion of local government’s river chief is mainly reflected in two aspects: The proportion of water pollution control indicators in various performance appraisal [37,72] and the assessment of water pollution control are regarded as important content of the “one vote veto” system in the promotion of the local government’s river chief [37].

Another important factor is the financial mechanism, which means that the main task of local governments is to improve the economy and strive for more taxes for local social and economic development. This ensures that the local government actively promotes large-scale projects that can generate tax revenue in a short period and ignores whether the environmental indicators meet the standards, so there is a situation of simultaneous treatment and pollution. The incentive mechanism can stimulate the potential of the local government’s river chief for water pollution control, while the financial mechanism has a certain reaction to the local government’s river chief for water pollution control. This kind of contradictory external pressure may cause the local government to adopt a symbolic or loose pollution control strategy in the process of implementing the RCP, thus stopping the actual effect of water pollution control to be up to the standard.

Therefore, how to adjust the efforts of local government river directors (i.e., to adjust the coefficient of reward, fine, and inferior) to make the RCP effective is a question that must be answered. According to the incentive theory, all employees want to be treated fairly, so the promotion incentive mechanism of the local government’s river chief should not be differentiated [73]. This means that the incentive mechanism cannot have a positive impact on the effect of water pollution control. Then, only under the different economic development level, the adjustment of the local government financial mechanism can obtain a better water pollution control effect and promote the effectiveness of the RCP’s implementation.

The central government’s annual assessment of the local governments includes two parts: economic and environmental assessments. If we adopt the same assessment standard for all regions of the country, it will inevitably lead to the underdeveloped regions being unable to complete the economic assessment task and environmental assessment task at the same time. Therefore, for underdeveloped areas in China, we should appropriately reduce their economic assessment tasks or give them more financial subsidies to help them better complete the annual assessment. However, for the developed areas in China, due to their rapid economic development and people’s higher requirements for a better living environment, they can rely on their own local finances to complete the assessment, without the need for additional financial subsidies from the central government. Therefore, reducing the economic assessment standard or giving extra financial subsidies to areas with slower economic development results in reducing the reward and punishment coefficient of water pollution control.

#### 5.3.3. Joint Impact of σ and θ on Water Pollution Control Effect of the RCP

Based on the analysis of Section 5.3.1 and Section 5.3.2, this paper found the four intervals constructed by σ and θ. Therefore, this section discusses the water pollution control effect in four situations.

(1)σ≤0.0403, θ<0.0063.

When σ≤0.0403 and θ<0.0063, there are a few random interference factors in the process of water pollution control, and as long as the local government carries out water pollution control, the water quality can be improved or the water quality itself is already very good, so there is no need to implement the RCP.

(2)σ≤0.0403, θ≥0.0063.

When σ≤0.0403, as long as θ≥0.0063, the RCP is effective in the Chaohu Lake basin. In addition, based on the analysis of Section 4.2.2, we know that the greater the rewarding excellence and punishing inferiority coefficient (θ), the better the water pollution control effect as a whole. However, at the same time, we know that the economy of the Chaohu Lake basin is relatively undeveloped; therefore, as long as the RCP is effective, we should relax θ appropriately.

(3)σ>0.0403, θ≥0.268.

When σ>0.0403, the effect of water pollution control fluctuates greatly, so it is necessary to ensure that θ≥ 0.268 to make the RCP effective. At the same time, we must also understand that the greater the random interference coefficient (σ), the greater the difficulty of governance. Therefore, on the one hand, it is necessary to increase the strength of reward, superior punishment, and inferior punishment to ensure the effectiveness of the RCP; on the other hand, it is also necessary to pay attention to the pressure on local governments [73]. The superior government should relax the assessment of a local government’s economic performance or should give financial support.

(4)σ>0.0403, θ<0.268.

When σ>0.0403, and θ<0.268, the RCP is invalid. The high random interference coefficient (σ) means that water pollution treatment is affected by more additional random interference factors, and the treatment is difficult. At this time, if the superior government only adopts some the rewarding excellence and punishing inferiority systems with no practical effects, such as oral warnings and meeting criticisms, then the RCP has no effect on water pollution control.

Therefore, in order to ensure the effectiveness of the RCP, it is necessary to ensure that σ ≤ 0.0403 and θ≥ 0.0063 or σ > 0.0403 and θ ≥ 0.268, to convert the theoretical results into ground implementation. For the random interference coefficient and the rewarding excellence and punishing inferiority coefficient, we can take certain measures into practice, and we can study the effectiveness of the RCP and thereby obtain the objective of water pollution control. Regarding the random interference coefficient, we can increase the amount of vegetation, control the amount and timing of fertilizer, or increase the penalties and control for the act of dumping sewage from enterprises. Regarding the rewarding excellence and punishing inferiority coefficient, we can improve the promotion mechanism and adjust the financial mechanisms for the river chiefs.

## 6. Conclusions

As a new water pollution control policy, the RCP has attracted widespread attention. Taking Chaohu Lake Basin as an example, we determined the function of the water pollution control effect function and influencing factors of COD in the Chaohu Lake Basin by building a differential game model; secondly, we compared the water pollution (COD) control effect of Chaohu Lake in the two cases, and then we analyzed the influence of random interference factors and the rewarding excellence and punishing inferiority coefficient (θ) on the water pollution (COD) control effect of Chaohu Lake. Finally, we obtained the parameter interval to ensure the effectiveness of the implementation of the RCP. The main conclusions are as follows:

(1)Generally speaking, compared with the non-RCP, the RCP improved the effect of water pollution control; therefore, the RCP should be used for water pollution control.(2)When implementing the RCP to control water pollution, the greater the rewarding excellence and punishing inferiority coefficient (θ) were, the better the water pollution control effect was.(3)The larger the value of the random interference coefficient (σ) was, the stronger the fluctuation of the water pollution control effect was.(4)To ensure that the RCP is effective, the random interference coefficient (σ) should be less than 0.0403, and the rewarding excellence and punishing inferiority coefficient (θ) should be larger than 0.0063; otherwise, when the random interference factor (σ) is greater than 0.0403, the rewarding excellence and punishing inferiority coefficient θ should be greater than 0.268.(5)When the random interference coefficient (σ) is large, this indicates that water pollution control is difficult for the local region. Therefore, on the basis of ensuring the effectiveness of the RCP, the superior government should appropriately adjust the assessment system of the local government river chief according to the economic development of different regions so as to relieve this pressure.

## Figures and Tables

**Figure 1 ijerph-17-03103-f001:**
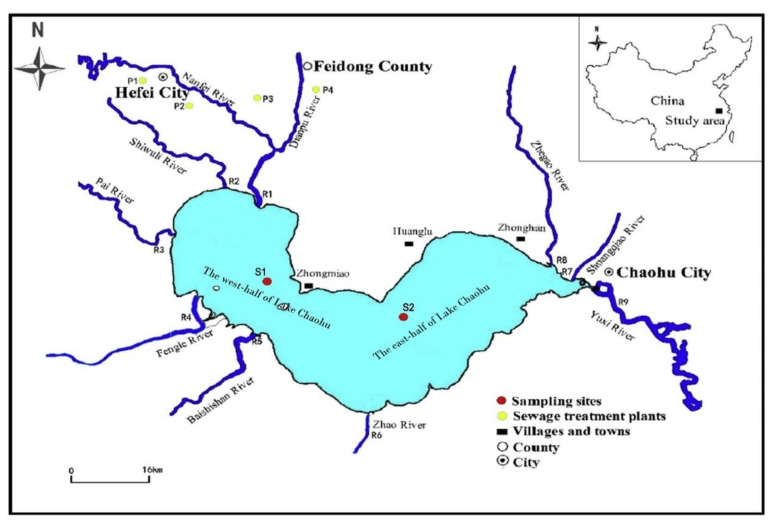
Study area of Chaohu Lake in China.

**Figure 2 ijerph-17-03103-f002:**
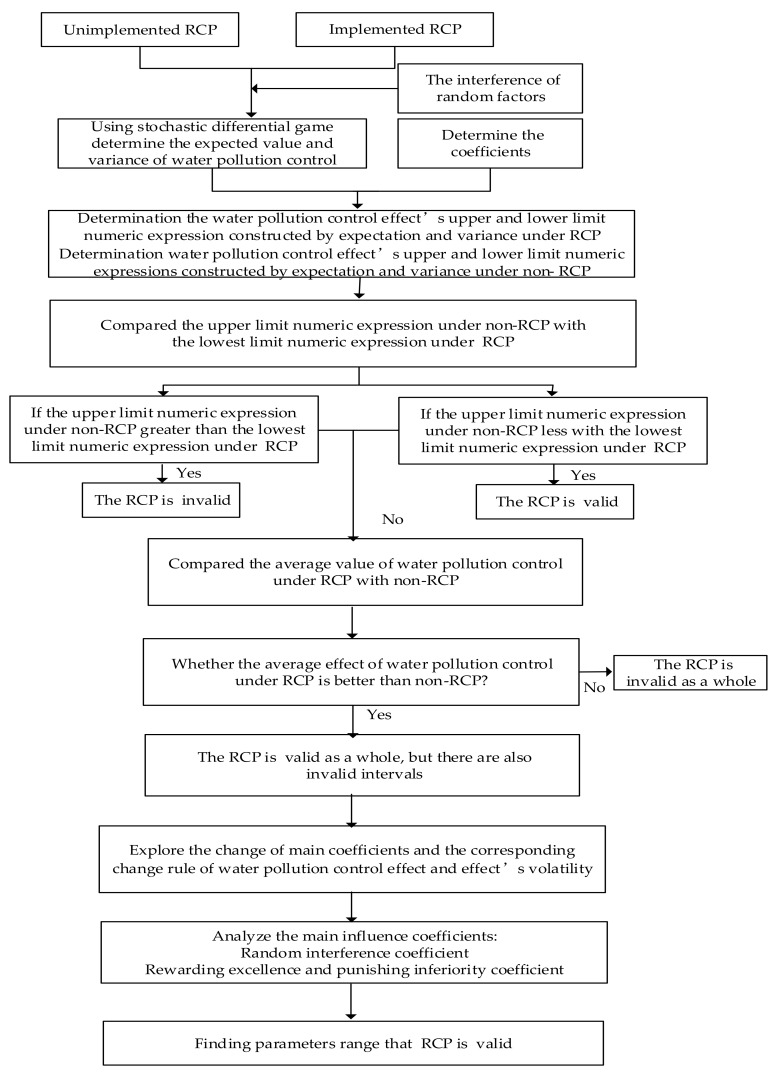
Research framework of this study on the River Chief Policy (RCP).

**Figure 3 ijerph-17-03103-f003:**
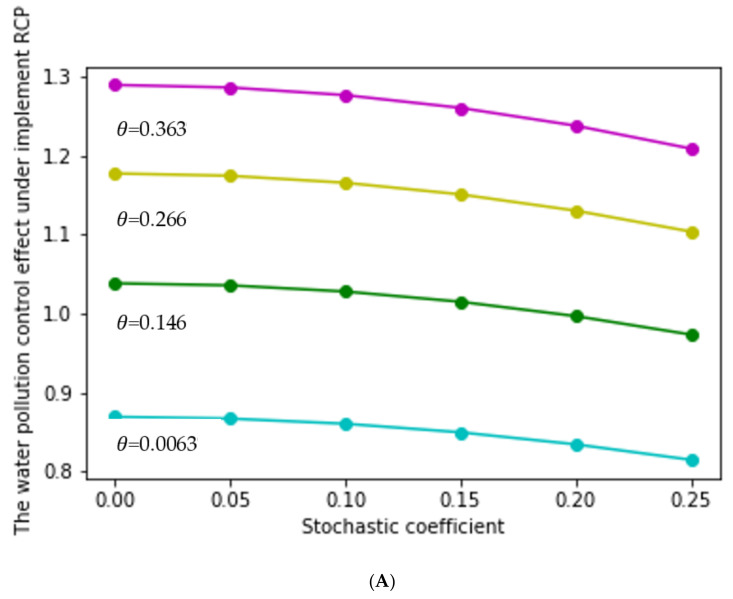
The RCP’s water pollution control effect limit values under different θ values. (**A**) The RCP’s water pollution control effect lower limit values under different θ values. (**B**) The RCP’s water pollution control effect upper limit values under different θ values.

**Figure 4 ijerph-17-03103-f004:**
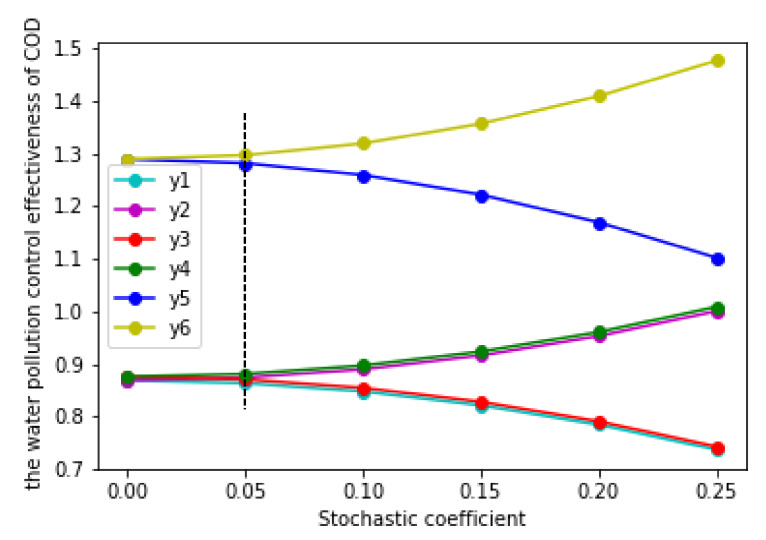
The water pollution control effect measured by in the two situations.

**Figure 5 ijerph-17-03103-f005:**
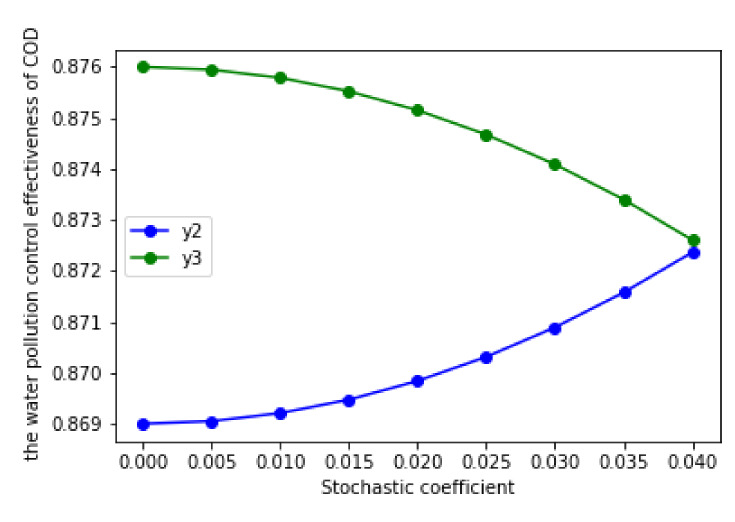
The interval of random interference coefficient (*σ*) to ensure the validity of the RCP.

**Figure 6 ijerph-17-03103-f006:**
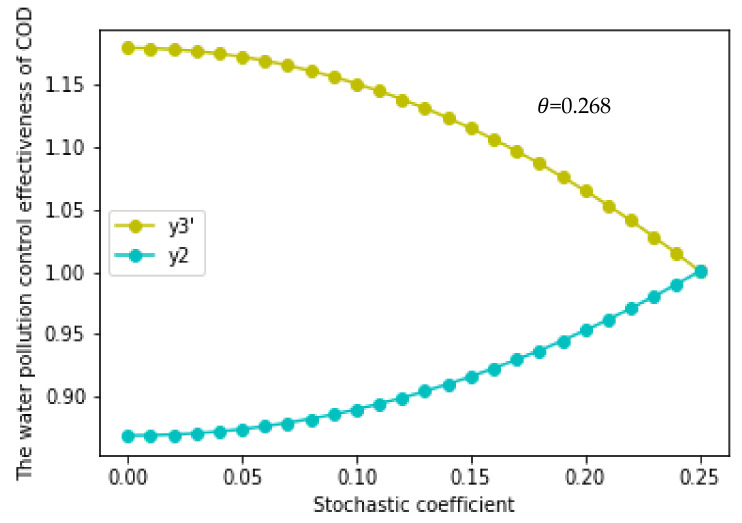
The interval of the reward and punishment coefficient (θ) to ensure the validity of the RCP.

**Table 1 ijerph-17-03103-t001:** The specifications of symbols used in this paper.

Symbols	Description
$y_{1}$	The lower limit function under the non-RCP
$y_{2}$	The upper limit function under the non-RCP
$y_{3}$	The minimum lower limit function under the RCP
$y_{4}$	The minimum upper limit function under the RCP
$y_{5}$	The maximum lower limit function under the RCP
$y_{6}$	The maximum upper limit function under the RCP

**Table 2 ijerph-17-03103-t002:** The coefficient values of Chaohu Lake in 2018.

Coefficient	Coefficient Value	Coefficient	Coefficient Value
δ	0.405	τ	0.985
α	0.333	μ	0.7
β	0.911	γ	0.005
θ	(0.0063, 0.363)	σ	(0, 0.25)

**Table 3 ijerph-17-03103-t003:** The numerical expression of water pollution control effect under the unimplemented RCP.

Content	Functional Expression	Numeric Expression
Unimplemented RCP	$y_{1}=E\left(p\left(t\right)\right)-1.96D\left(p\left(t\right)\right)$	$y_{1}=0.869-2.103\sigma^{2}$
	$y_{2}=E\left(p\left(t\right)\right)+1.96D\left(p\left(t\right)\right)$	$y_{2}=0.869+2.103\sigma^{2}$

**Table 4 ijerph-17-03103-t004:** The water pollution control effect under the implemented RCP.

Content	Functional Expression	Numeric Expression
Implemented RCP θ=0.0063	$y_{3}=E^{h}\left(p\left(t\right)\right)-1.96D^{h}\left(p\left(t\right)\right)$	$y_{3}=0.876-2.121\sigma^{2}$
	$y_{4}=E^{h}\left(p\left(t\right)\right)+1.96D^{h}\left(p\left(t\right)\right)$	$y_{4}=0.876+2.121\sigma^{2}$
Implemented RCP	$y_{5}=E^{h}\left(p\left(t\right)\right)-1.96D^{h}\left(p\left(t\right)\right)$	$y_{5}=1.289-3.119\sigma^{2}$
θ=0.363	$y_{6}=E^{h}\left(p\left(t\right)\right)+1.96D^{h}\left(p\left(t\right)\right)$	$y_{6}=1.289+3.119\sigma^{2}$

**Table 5 ijerph-17-03103-t005:** The average value of water pollution control effect in two situations.

Content	The Mean Value
Unimplemented RCP	0.869
Implemented RCP (θ=0.0063)	0.876

**Table 6 ijerph-17-03103-t006:** The water pollution control effect and the effect’s volatility.

Symbols	Water Pollution Control Effect Value	The Effect’s Volatility
$y_{1}$	[0.737,0.869]	[0,0.135]
$y_{2}$	[0.869,0.872]	
$y_{3}$	[0.743,0.876]	[0,0.266]
$y_{4}$	[0.876,1.009]	
$y_{5}$	[1.094,1.289]	[0,0.390]
$y_{6}$	[1.289,1.484]	

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
