# Peer review of "Assessing the Effect of the Chinese River Chief Policy for Water Pollution Control under Uncertainty—Using Chaohu Lake as a Case"

_ijerph, 2020, doi:10.3390/ijerph17093103_

Round 1
Reviewer 1 Report
The topic addressed is of great interest in all latitudes and develops an interesting method that through a differential model allows to distinguish the impacts of an environmental policy applied to the respective body of water and the incidence of random interference on water courses.
The manuscript is very extensive, it requires a structural revision that clarifies the information delivered in a better way.
Relevant concepts such as punishment and incentive for excellence that must be defined more clearly.
A more detailed explanation of numerical variables, mathematical development, and "associated physical" concepts.
This also applies to the political and administrative aspects, which are included.
I recommend a table for the previous suggestions, in such a way that it is easier to read and conceptual link the work.
An aspect that calls attention that all the exposed graphs are perfect curves, in such a way that the random interference is not expressed.
Finally I would recommend that an effort be made to be shorter.
Author Response
Dear Reviewer:
Thanks very much for your kind letter about the proof of our paper titled “Assess the effect of Chinese River Chief Policy for Water Pollution Control Under Uncertainty—Using Chaohu Lake as a Case” for publication in “International Journal of Environmental Research and Public Health”. We have finished the proof reading and checking carefully, and some corrections about the proof and the answers to the queries are provided below.
1.The manuscript is very extensive, it requires a structural revision that clarifies the information delivered in a better way.
Thank you for your advice and suggestions. According to your opinion, the rest of this paper is arranged as follows: section 2 describes case study and data sources; section 3 introduces methods; section 4 provides the main results; section 5 provides the main discussion; section 6 presents the conclusion.
Specifically, we have adjusted the details of section 2 and section 3. Please see section 2 and section 3, line 119 to 312.
- Relevant concepts such as punishment and incentive for excellence that must be defined more clearly.
Thank you for the expert's advice. According to your opinion, we define the parameters more clearly. We have explained the rewarding excellence and punishing inferiority, which is shown as “Specifically, each body of water (divided by administrative area) is managed by special local officials, and they are assessed by their superiors based on the improvement of water quality. If the local government's actual water pollution treatment effect meets the national standard, the state will reward the local government's river chief; otherwise, if the local government's actual water pollution treatment effect does not meet the national standard, the state will punish the local government's river chief.” Please see the line 57-62.
For similar situations in this paper, we explain other concepts: the effect of water pollution control and new institutional economics more clearly.
We have explained the effect of water pollution control means the reduction of water pollutions. We revise the paper as follows:” the effect of water pollution control (the reduction of water pollutions).” Please see the line 82.
We correct the new institutional economics into the transaction cost, property right, and institutional change theory. We revise this sentence as follows: “Wang and Cai analyzed the advantages and disadvantages of RCP from transaction cost, property right, and institutional change theory.” Please see the line 72.
3.A more detailed explanation of numerical variables, mathematical development, and "associated physical" concepts. I recommend a table for the previous suggestions, in such a way that it is easier to read and conceptual link the work.
Thank you for the expert's advice. According to your opinion, we explain the main variables in this paper and present them in tables, just as follows. Please see Table A4 in Appendix A, line 623.
Table A4. The main Variables and descriptions
|
Variables |
Description |
|
E(p(t)) |
It means the expectation of the effectiveness of water pollution control under non- RCP. It expresses the average value of water pollution control effect under non- RCP. It reflects the trend of the effect of water pollution control’s data centralization. |
|
D(p(t)) |
It means the variances of the effectiveness of water pollution control under non- RCP. It reflects the degree of random variable deviates from its expectation under non-RCP. It measures the volatility of water pollution control effects of non-RCP. |
|
|
It means the expectations of the effectiveness of water pollution control under RCP. It expresses the average value of water pollution control effect under RCP. It reflects the trend of the effect of water pollution control’s data centralization under RCP. |
|
|
It means the variances of the effectiveness of water pollution control under RCP. It reflects the degree of random variable deviates from its expectation under RCP. It measures the volatility of water pollution control effects under RCP. |
4.This also applies to the political and administrative aspects, which are included. I recommend a table for the previous suggestions, in such a way that it is easier to read and conceptual link the work.
Thank you for the expert's advice. According to your opinion, we describe the policy and administration of RCP more clearly. Please see Table A5 in Appendix A, line 624.
Table A5. Policy and descriptions
|
Police |
|
Description |
|
RCP
|
Meaning |
Government officials will be hired as river chief. Their responsibilities will include water resource protection, pollution prevention and control, and ecological restoration. Their job performance will be assessed and they will be held accountable if environmental damage occurs in the water they take charge of. |
|
Development |
In 2007, Wuxi City in Jiangsu Province first introduced and implemented RCP to address a blue algae outbreak in Taihu Lake. After RCP implemented, the water environment of Taihu Lake has rapidly improved. In 2008, the RCP was applied to several other cities in Jiangsu Province alongside Taihu Lake. From 2008 to 2016, RCP gradually expanded to Hebei, Yunnan, Hubei, Anhui and other provinces. In 2016, the General Office of the State Council officially published Comments claimed that the RCP would be implemented throughout the whole country by the end of 2018. |
5.An aspect that calls attention that all the exposed graphs are perfect curves, in such a way that the random interference is not expressed.
Thank you for the expert's advice. This paper all the exposed graphs are perfect curves exactly, while they can represent randomness. Besides, according to your opinion, in order to show the randomness obviously, we add Figure A1 in Appendix A, line 625 to express the random interference of the effect of water pollution control and we explain it in the line 234 to 237. Form Figure A1, we can see the effect of water pollution control around its expected value fluctuates under different deserved periods. It implied that the value of the water pollution control effect is uncertainty. That is to see, it is impossible to obtain the exact value of the water pollution control effect. Therefore, we cannot to compare the effect of RCP with non-RCP by fixed water pollution control effect value, and the value of the water pollution control effect should be characterized by a confidence interval.
Besides, this paper’s all the exposed graphs are perfect curves have two reasons. Firstly, the randomness mainly affects the random coefficient in the model, and the final express form of this model is an interval with the random coefficient. At the same time, we analyze and discuss the interval about random coefficient. Secondly, the actual effect of water pollution control is fluctuated between E(p(t))-1.96D(p(t)), E(p(t))+1.96D(p(t)), or . While the upper and lower limits are changing in the interval, so we focus on to analyze the two extreme cases. Besides, by calculation, we find that the upper and lower limit functions are the quadratic function about the random interference coefficient. Therefore, this paper’s all the exposed graphs are perfect curves.
Figure A1 is added in the modified attachment.
- Finally, I would recommend that an effort be made to be shorter.
Thank you for the expert's advice. According to your suggestion, we have properly simplified the introduction and method. Please see the line 71-72, line 82-92, and line 241-252.
Finally, thanks again for the expert’s review and efforts!

Reviewer 2 Report
The authors present a model to examine the effectiveness of the River Chief Policy on Lake Chaohu in China. I was not able to give a complete review, I am not familiar with the modelling approach discussed. One question comes to mind as to whether the modelling approach is required, or if monitoring and social data can be used to examine before and after effects of the policy. I don't see a clear purpose for the model and since a large portion of the paper discusses model development, maybe it would be ore useful for another journal such as Ecological Modelling. Some more comments appear below.
Line 21 – “water pollution control effect” Does this mean a reduction in water pollution? Please modify wording to make this clearer.
Lines 22-24 – This sentence makes it sound as if adjusting the coefficients will reduce water pollution. Maybe in the differential game model but not on the ground. Perhaps add in the model somewhere in this sentence to clarify that. How does this relate to on the ground implementation?
The aims of this study, as introduced in the introduction, are somewhat confusing. Is it to assess the effectiveness of the RCP or look at “interference of random factors”? The review of the literature is rather linear and not synthesized. The authors simply list a series of studies, but the relationship to the aims is not always clear. It appears that the study will focus on the introduction of random factors. This should probably be the focus of the study, using the RCP as a case study. The introduction should be rewritten and focus on this. The case study information should be moved to a separate section. This will give the manuscript a broader more international focus. The sentence on line 112 encapsulates the aim of the study to look at random factors. This should be the main aim that could be applicable to policies around the world.
Line 37 – reference 12 refers to a study in Brazil, not the United States.
Line 46 – punishment is not the right word here. Perhaps change this to “penalties” or “fines” or “fees” or “sanctions”
Line 66 – What are “new institutional economics”?
Line 69 – What is meant by “put forward the corresponding optimal paths”? What are optimal paths?
Line 73 – What are these? “supervision means, index construction and assessment method” These descriptions are too vague.
The methods section has poor English language and much repetition. The errors are too numerous and should be rewritten under the guidance of an English language expert. The date were collected from the Ministry of Ecology and Environment, but there is no information about when the RCP policy was implemented in relation to this data. Table A2 was not provided in the manuscript downloaded from the MDPI website.
Line 120 – “the stochastic differential game” Is this an established method? A refeence should be provided here.
Line 151 – COD’s are not previously defined. What are they?
I am unable to comment properly on the model. Interesting approach, but I am not really sure that a model is needed to see if a policy is effective or not. Isn’t the lake monitored and would you be able to see the effectiveness in the monitoring data? What abut social and health aspects. Surely, you could examine water usage, illnesses and other factors to assess the effectiveness. The role of the model in the in achieving the research outcomes needs to be more clearly defined and explained.
Some references (e.g. # 14) are not properly formatted. Please check reference formatting.
Author Response
Dear Reviewer:
Thank you for your review. we have revised one by one as required.Please see the attachment for details.

Round 2
Reviewer 1 Report
I appreciate the willingness on your part to the suggestions made and your work will be an important reference for those of us who work linked to the resource, water, in its different dimensions.Author Response
Dear Reviewer: Thank you for your affirmation of our research!Reviewer 2 Report
The authors have addressed the main comments from the first review. However, the manuscript still requires extensive English language editing. I have provided some edits for the first few sections. The authors show seek out some assistance in editing the remaining sections of the manuscript and resubmit. There are also some comments below which the authors should address. Primarily, the authors should attempt to highlight the global relevance of their work and integrate their research water pollution control measures in a more holistic sense.
Abstract
Line 13 – “dealt with the” should be “manage”
Line 14 – What sort of basins? Catchments? Watersheds? River basins?
Line 14 – “ the system” should be “policy”
Line 14 – remove “therefore”
Line 15 – “the water pollution of Chaohu Lake” should be “water pollution in Chaohu Lake”
Line 17 – “The” should not be capitalized
Introduction
Line 33 – “to deal with it” should be “to manage the problem”
Line 34 – “framework” should be “frameworks”
Line 37 – “because it cannot achieve the maximization of” should be “due to its inability of maximizing”
Line 40 – “Compared with developed countries, due to the weak institutional background, the developing countries’ decentralized water environment policy is not as successful as in the United States[10].” I would rewrite this…” Due weak institutional structure decentralized water policy in developing nations is less successful than in the United States”
Line 48 – “It caused that the effects of some measures taken by governments at all levels in China was not significant.” Should be “This has resulted in ineffective pollution control measures across all levels of [government?] in China.
Line 49 – remove “an”
Line 62 – reward and punish – How will they be rewarded and punished? Include some examples.
Line 64 – add “is” between “water quality” and “linked” and “a” between “with” and “local”
Line 74 – Are these separate paths? Then the sentence should be rewritten as follows…“rule of law, rule of virtue and autonomy” to optimize the RCP, respectively.”
Line 76 “and gave some policy” should be “and have provided”
Line 77 should “supervise” be “supervisory”
Line 79 – “These studies can help us to know what is RCP, its path to governance water pollution and how to assess, but none of them evaluate and analyze the main coefficients to ensure the effect of RCP from a quantitative perspective.” This sentence needs to be rewritten. Here is a suggestion, “These studies can help us understand the what is the RCP is, its path to governance and how to assess it, however the is a lack in understanding of how to evaluate and analyze the main coefficients or factors that ensure effectiveness of the RCP from a quantitative perspective.” I would then go on to say why you think this is important, to understand the quantitative perspective.
Line 83 – presents uncertainties due to stochasticity, such as
Line 84 – illegally discharged sewage
Line 84-85 – delete “, which has affected the effects of water pollution control”
Line 85-87 – I don’t understand this sentence
Line 88, remove “but”
Line 87-92 – rewrite this sentence -“While, many scholars have studied the related problems of water pollution control, but they have determined the effect of water pollution control without considering the interference of random factors, and the effect of water pollution control is fixed value, which caused some errors [33,37-42]. Therefore, in this paper, we are integrated random interference factors into the evaluation of water pollution control effect under the cases of implementing RCP and not implementing RCP, and then evaluate RCP’s results more accurate.”
Here is a suggestion…
“While many scholars have studied the problems related to water pollution control, they have failed to consider the interference of random factors (stochastic events). This paper assesses the integration of random interference factors in evaluating the effectiveness of water pollution control both in the presence and absence of the RCP.”
Line 93 – How is this relevant to the broader, global knowledge of water pollution control? This paragraph needs to be rewritten. There is poor English language here and this makes it difficult to understand.
Line 94 – Why does is matter that there is no quantitative study?
Line 107 – The paper should have broader, global implications. Those should be stated here. How does your quantitative model assessing random interference factors improve not only the RCP but pollution control measures around the world.
Line 118 should be “Data Sources”
Study Area and Data Sources
Chaohu Lake and man-made lake? If so, does the model consider increasing flow from the inputs and outlets to dilute the pollution?
The remaining sections should be edited, extensively, for the English language corrections.
Author Response
Dear Reviewer:
Thanks very much for your kind letter about the proof of our paper titled “Assessing the effect of the Chinese River Chief Policy for Water Pollution Control Under Uncertainty—Using Chaohu Lake as a Case” for publication in “International Journal of Environmental Research and Public Health”. We have finished the proof reading and checking carefully, and some corrections about the proof and the answers to the queries are provided PDf file.
